# Standard automated perimetry using size III and size V stimuli in advanced stage glaucoma: an observational cross-sectional comparative study

Devindra Sood,[1,2,3] Gabriela Czanner,[4] Tobi Somerville,[5] Ishaana Sood,[6] Fiona J Rowe [7]

For numbered affiliations see end of article.

**Correspondence to**
Prof Fiona J Rowe;
rowef@liverpool.ac.uk

## ABSTRACT

**Objectives** In this study, we sought to evaluate the extent of further visual field that could be assessed when using stimulus size V in standard automated perimetry compared with size III in advanced stage glaucoma and whether cut-off values could be determined for when to switch from size III to size V.

**Design** Prospective cross-sectional study.

**Setting** Single-centre outpatient eye clinic in India (New Delhi).

**Participants** Advanced stage glaucoma defined as stages 3–4.

**Intervention** Central static perimetry with Octopus 900 G programme (size III stimulus dynamic strategy) and low vision central programme (size V stimulus dynamic strategy).

**Primary and secondary outcome measures** Visual field assessment for right and left eyes with both sizes III and V were undertaken within one clinic visit.

**Results** We recruited 126 patients (170 eyes). Mean patient age at assessment was 55.86 years (SD 15.15). Means (SD) for size III versus size V, respectively, were 6.94 dB (5.58) and 12.98 dB (7.77) for mean sensitivity, 20.02 dB (5.67) and 19.22 dB (7.74) for mean deviation, 5.89 dB (2.29) and 7.69 dB (2.78) for standard loss variance and 3.32 min (1.07) and 6.40 min (1.43) for test duration. All except mean deviation were significantly different between size III and V tests.

**Conclusion** Useful visual field information was obtained with size V stimuli which allowed continued monitoring of these patients that was not possible with size III. Increased test duration, standard loss variance and mean sensitivity were found with size V, as expected, given that more visual responses were obtained with the increased target size. A switch from size III to V may be considered when mean sensitivity reaches 10 dB and/or mean deviation reaches 18 dB.

## STRENGTHS AND LIMITATIONS OF THIS STUDY

⇒ Size V stimulus in standard automated perimetry has a greater dynamic range than size III stimulus.

⇒ Visual field results of advanced stages 3–4 glaucoma meeting reliability criteria were analysed.

⇒ Sample size calculations were met for recruitment to this study.

## INTRODUCTION

Standard automated perimetry using stimulus size III is an internationally accepted visual field assessment for glaucoma.[1] Static threshold perimetry is particularly adept in aiding detection of early visual field loss in glaucoma. For patients with glaucoma, the usual choice of visual field assessment is static perimetry using a threshold strategy to assess the central 24°–30° of the visual field of right and left eyes individually.[1] In clinical practice, a number of issues can arise with these central assessment programmes. As glaucoma progresses and the central visual field deteriorates to the extent that minimal (if any) visual field remains, appropriate and reliable evaluation can become difficult.

For patients with severe to advanced stage disease, central visual field assessment can show substantial visual field loss and, therefore, is limited in providing useful information about further progression or stability of the visual field loss, or for providing information about the individual's functional visual status. In these patients, further analysis options for visual field assessment must be considered. Where the central visual field island remains, it is possible to target assessment to the central 10 degrees of visual field. Tomairek *et al*[2] showed a considerable number of eyes with severe glaucoma with absolute visual field defects on 24–2 programme testing showing relative defects involving only some points of the central 10° on the 10–2 programme. Thus, they stressed the importance of detailed central testing to underpin intensive treatment to preserve the residual central visual field.

A further alternative is to continue assessment of the central 30° but to increase the

stimulus to size V.[3–5] While there is considerable debate over the use of different stimulus sizes in automated perimetry,[6–10] from a clinical perspective, size III remains the standard stimulus in mainstream clinical use.[1 7] However, when visual field results using standard automated perimetry show severe loss, use of size V stimulus becomes a consideration.

There are conflicting reports of size V vs III in early visual field loss. The use of the larger stimulus size could prevent the detection of small localised defects because the large stimulus size stimulates not only the area of visual field loss but also the surrounding, better, areas of visual field.[6] However, Wall et al[5] found that when the data were binned by mean deviation, size V maintained its advantage to detect visual loss even in the bin with the best mean deviation. Further, Flanagan et al[11] reported that while size III testing resulted in a greater depth of the defect, size V, because of its better retest variability, flagged abnormal test locations in early glaucoma at least as well as size III. This is not directly relevant to studies that focus specifically on severe visual field loss. With severe visual field loss, when using size III, it is not possible to gather reliable visual field data by increasing stimulus contrast (ie, lower decibel values) as often there is no response within damaged areas of visual field.[6 7] Thus, a further option is to increase the target size instead.

The useful dynamic range of perimetry is the range of disease severities over which useful measurements can be obtained. With increasing visual field loss, the response within the area of visual field loss becomes more variable because of decreased retinal ganglion cell response rate along with the limited number of stimulus presentations in each location of the test programme.[6 7] Use of size V stimulus has been shown to have a greater dynamic range and a similar number of abnormal test locations when compared with the traditionally used size III stimulus.[3] Higher sensitivity is achieved at the same location giving more reliable and less variable estimates of sensitivity at the damaged visual field locations. Gardiner et al reported that for rapidly progressing eyes, use of size V could represent more than 5 years of added, useful and reliable test results. For less rapid loss, size V could provide many years of reliable clinical information.[6]

In this study, we aimed, in a pragmatic clinical study, to investigate the information obtained from central static visual field assessment comparing size V vs size III targets using the Octopus 900 G and low vision central (LVC) programmes in patients with advanced stage glaucoma. We hypothesise that, for many patients, we would obtain useful visual field results with size V to use as a new baseline for visual field status for subsequent comparison over follow-up, and further, comparative data of size III and V could be used to provide a recommendation for transfer from one test strategy to the other.

## MATERIALS AND METHODS
### Design
We undertook a prospective, comparative, observational cross-sectional study in a single centre in New Delhi, India. We followed the Strengthening the Reporting of Observational Studies in Epidemiology checklist for reporting cross-sectional studies.

### Recruitment
The study recruited patients with visual field loss due to advanced glaucoma and who met our inclusion criteria. Consecutive patients with an existent clinical diagnosis of glaucoma requiring visual fields were recruited from ophthalmology outpatient clinics between 2016 and 2017. Severe glaucoma was diagnosed according to the International Classification of Diseases (ICD) diagnostic code 9 (365.73) and 10 (7th digit '3'). Diagnostic code 365.73 represents the glaucoma stage code for severe, advanced and end-stage glaucoma consisting of 'glaucomatous visual field abnormalities in both hemifields and/or loss within 5° of fixation in at least one hemifield'.[12 13] ICD version 9 codes were updated in 2015 to version 10 codes which consist of seven digits.[14] The first three indicate the code category and the last four provide added detail. Where the last digit is '3', this indicates severe glaucoma regardless of the type or cause of glaucoma. For example, H4010×3 represents unspecified open angle severe stage glaucoma.

### Inclusion criteria
We included adult patients attending for visual field assessment with a diagnosis of severe or advanced glaucoma (stages 3–4), sufficient motor ability to sit at the perimeter unaided, ability to press the response button, sufficient cognitive ability to understand and follow instructions for performing the test, and willingness to undergo standard visual field assessments on the same day.

### Exclusion criteria
We excluded patients with a diagnosis of stages 1–2 glaucoma, visual acuity worse than 1.0 logMAR (logarithm of the Minimal Angle of Resolution), those unable to sit at the perimeter, those with unreliable visual fields, unable to follow instructions for performing the test or too ill to complete the full assessment.

### Patient and public involvement
This study addresses a top research priority identified by patients and the public in a national (UK) consultation process: 'what is the most effective way of monitoring the progression of glaucoma?' (https://www.jla.nihr.ac.uk/). Patients were not involved directly in the design and conception of this study.

### Visual field assessment
Past studies have frequently used the 24–2 programme when testing visual fields in advanced glaucoma.[3–5] The Octopus 900 perimeter (Haag Streit AG, Bern, Switzerland) G programme (dynamic strategy) has a

physiology-based grid of 59 test locations within the central 30°.[15] Locations are clustered more closely together centrally (2.8° spacing) with five central foveal locations and 17 test locations in the macular region. Locations are spaced further apart peripherally with emphasis on locations in nasal step regions and with more test locations nasally than temporally. There is no weighted analysis of test locations. Test locations are distributed in a pattern to follow retinal nerve fibre bundles to facilitate the detection of glaucomatous visual field loss. The Octopus 900 perimeter LVC programme (dynamic strategy) has a grid of 75 test locations within the central 30°.[15] One location is within the central fixation area and the remaining 74 locations are spaced equally apart by 6° with an off-set from the horizontal and vertical meridians of 2°. Static visual field results were deemed unreliable if combined false negative and false positive responses exceeded 25%.

The study protocol consisted of visual field assessment with static G Octopus 900 perimetry using size III and static LVC size V targets (two tests per eye) on the same day. The order of assessment was randomised for size III vs size V. The G and LVC programmes were run as standard with the size III target size (0.43° in diameter) or with the size V target size (1.72° in diameter).

## Sample size

We studied two main outcomes: mean sensitivity and mean deviation.

For mean sensitivity sample size, we assumed a difference of 3 dB to be of clinical importance. Wall *et al*[5] reported lower and upper quartiles of 17.4 dB and 24.5 dB which are equivalent to an SD of 3.55. Based on these figures, with pairwise t-test, significance level of 2.5% and power of 85%, this leads to a sample size of 15 eyes (1 eye per patient).

For mean deviation, in the sample size we assume a difference of 10% decrease to be clinically relevant. Wall *et al*[4] showed a 15% decrease in their patients, while pooled SD was 5.5 dB.[7] With pairwise t-test, and significance level of 2.5%, and power of 85%, this leads to a sample size of 126 eyes (1 eye per patient).

## Statistical analysis

A direct comparison of results was made for static results using the statistical package SPSS V.25 (IBM SPSS Statistics). The unit of analysis was per eye.[16]

Glaucoma typically affects both eyes, thus, each eye is not considered independent when considering analysis of visual field assessments. Assessments are usually reported for worst affected eye only. In this study, we sought to evaluate visual field data available from size V vs size III perimetry. We had all the data from both eyes for many patients, hence we aimed to use all the data from all eyes as our primary analysis. For the purposes of sensitivity analysis, we also present results from worst eyes only.[16] We chose to include worst eyes only for this analysis on the basis that we wished to determine at what point within worst fields we should transfer from size III to size V testing. Worst eye

was defined as the eye with greater mean deviation value on G1 static perimetry.

To evaluate normality of distribution of results from right and left eyes, a Kolmogorov-Smirnov test was used. Further we conducted a general linear model analysis for difference between eyes in mean sensitivity and mean deviation. For this model, the difference in mean sensitivity, or mean deviation, value was the dependent variable and the eye (right vs left nested within the patient) was treated as random factor. We also did analysis of the visual field measurement difference in the worse eye via a paired t-test with Bonferroni adjustment for mean sensitivity and mean deviation calculations.

To compare the results of the G programme when undertaken with size III and size V targets, variation was evaluated for:

► Mean sensitivity.
► Mean deviation.
► Standard loss variance.

We further compared mean sensitivity and mean deviation values for each of the four quadrants of the visual field to determine if different quadrants showed comparable changes. Quadrants were defined as Q1 (superior nasal field), Q2 inferior nasal field, Q3 (superior temporal field) and Q4 (inferior temporal field) for both right and left visual field results. Duration of assessments was also compared.

Scatterplots include linear regression lines and locally estimated scatterplot smoothing (loess) curves. The linear regression line assesses strength of linear association between two continuous variables with $r^2=0.0$ indicating lack of linear association (ie, lack of correlation). Correlation is first assessed by its p value (p) and then, if significant, the correlation is judged as none, small, medium or large if $r^2$ is 0–0.01, 0.01–0.09, 0.09–0.25 or more than 0.25.[17 18] A non-directional p value, that is, two-sided alternative hypothesis, is used. The value of $r^2$ tells us how much variability is shared between variables. The loess curve is a local non-linear regression curve in which the fitting of each point is weighted towards the data nearest to that point. It makes no assumption about the association between variables and is used to visually observe the possible nature of the association between two continuous variables. It can be used as a modelling tool and provides a nonparametric regression that focuses on the fitted curve.[19] The fitted points and related standard errors do respect a particular estimate but are estimated to the whole curve.[20] The default span was set to values ranging from 0.60 to 0.80 as a trade-off to ensure sufficient data for an accurate fit in order to reduce variance, and to avoid an oversmoothed regression in order to reduce bias. Along the loess curve, the cross-section between the x and y axis relating to the main point of inflection along the curve was used to indicate cut-off values between data represented along the x and y axes, and therefore, for this study, used to calculate potential cut-off values for transfer of testing from size III to size V stimulus. The inflection point was taken to indicate at

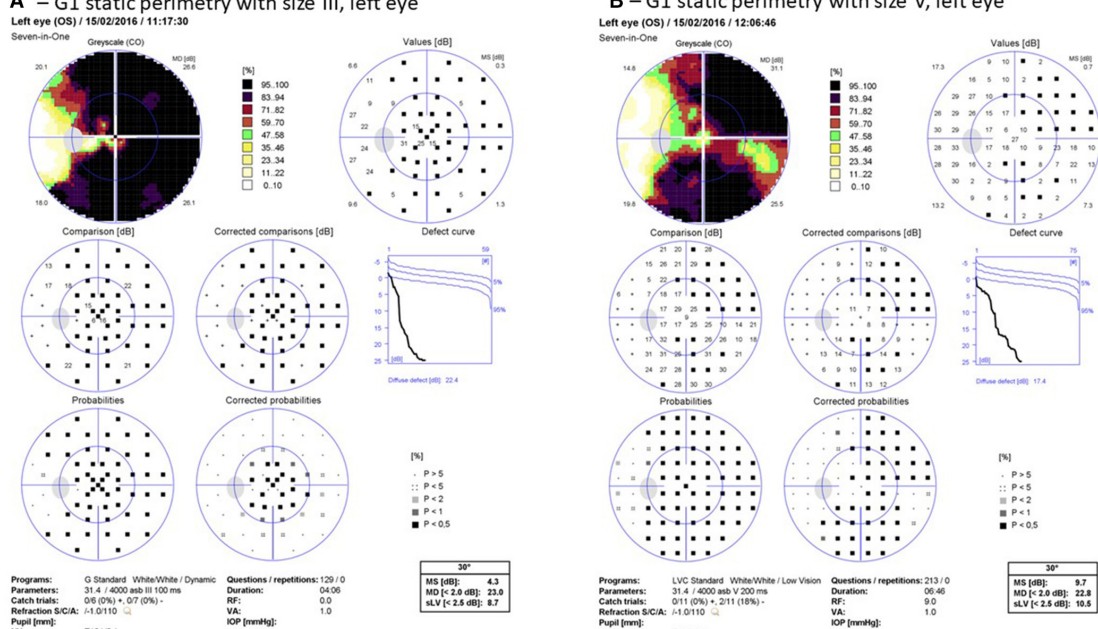

**Figure 1** Illustrative results of static size III versus size V perimetry. (A) G1 static perimetry with size III, left eye. (B) LVC static perimetry with size V, left eye. Static perimetry (using size III) results show a severely restricted visual field while static perimetry (using size V) results display better temporal visual field and inferonasal visual field. LVC, low vision central; MD, mean deviation; MS, mean sensitivity; IOP, intra ocular pressure; OS, left eye; RF, reliability factor; VA, visual acuity.

what value the relationship between III and V was possibly changing (as relevant when the linear relationship does not explain the association between III and V). The point of greatest curvature is suggested as a potential point of transitioning from size III to V as that is the point when the size III stimulus accelerates in its inability to detect change in visual field—relatively when compared with size V test. We report the inflection point as a range of values, as observed for the range of ($\alpha$) 0.60–0.80.

In this paper, all global indices are reported as positive numerical values, including mean sensitivity, mean deviation and standard loss variance.

## RESULTS
### Primary analysis: all patients/eyes
Results are presented for 126 patients (170 eyes) with reliable visual field performance; 29 eyes at stage 3 and 141 at stage 4. A further 46 eyes were excluded because of unreliable visual field performance or visual field results graded at stage 2 or better. The worst affected eye was right in 64 (50.8%) and left in 62 (49.2%). Mean age at time of testing was 55.86 years (SD 15.15; 58 (range 13–83)). Figure 1 displays one example of size III vs size V results for one patient.

### Distribution of data
Visual field assessment data was tested to be normally distributed (Kolmogorov-Smirnov test). Further, general linear model analysis with random effect being the subject and fixed effect being (1) eye laterality (right vs left eye) or (2) worst versus better eye, with respect to mean

sensitivity, showed no significant difference between size V versus size III data (p=0.888 and p=0.644 respectively).

### Size III versus size V differences
Results are outlined in table 1 for differences, and significance, for mean sensitivity, mean deviation and standard loss variance, along with differences across quadrants plus test duration. Differences for mean sensitivity between sizes III and V were about 6 dB overall (higher values for size V) and 5–7 dB across quadrants with greatest increases in inferior quadrants. A strong linear relationship and large correlation was found between mean sensitivity for size III and V stimuli (figure 2: $r^2$=0.731, p=0.0001). Using the loess curve to consider differences between size III and V stimuli, the point of greatest inflection (visualised on the scatterplot with value taken from y axis) is approximately 4–6 dB for size III stimulus.

Global mean deviation was significantly lower for size V but not clinically significant with significant differences only in inferior quadrants. A strong linear relationship and large correlation was found between mean deviation for size III and V stimuli (figure 2: $r^2$=0.735, p=0.0001). Using the loess curve to consider differences between size III and V stimuli, the point of greatest inflection is approximately 22–24 dB for size III stimulus. Diffuse defect was lower and standard loss variance was higher for size V (see online supplemental figure 1) representing more visual field responses with the size V target. Test duration increased for size V but was likely due to more patient responses being obtained due to presence of more visible targets.

**Table 1** Differences in visual field parameters for target size—all eyes

| | G1, size III | | G1, size V | | Difference | | General linear mixed model (subject) | Correlation significance | Bonferroni correction |
|---|---|---|---|---|---|---|---|---|---|
| | Mean | SD | Mean | SD | Mean | SD | | | |
| MS | 6.94 | 5.58 | 12.98 | 7.77 | −6.04 | 4.18 | 0.0001 | 0.005 | 0.0002 |
| MD | 20.04 | 5.67 | 19.22 | 7.74 | 0.82 | 4.15 | 0.023 | 0.007 | 0.046 |
| SLV | 5.89 | 2.29 | 7.69 | 2.78 | −1.80 | 2.60 | 0.0001 | 0.0001 | * |
| DD | 17.13 | 7.86 | 14.76 | 9.47 | 2.37 | 5.46 | 0.0001 | 0.009 | * |
| MS Q1 | 5.31 | 5.48 | 10.48 | 8.78 | −5.17 | 5.21 | 0.0001 | 0.0001 | * |
| MS Q2 | 7.70 | 7.48 | 14.40 | 9.74 | −6.70 | 5.33 | 0.0001 | 0.004 | * |
| MS Q3 | 6.27 | 5.78 | 12.05 | 8.68 | −5.77 | 5.42 | 0.0001 | 0.114 | * |
| MSQ4 | 8.23 | 7.59 | 14.68 | 10.12 | −6.45 | 5.37 | 0.0001 | 0.005 | * |
| MD Q1 | 21.29 | 5.55 | 21.13 | 8.70 | 0.16 | 5.16 | 0.662 | 0.001 | * |
| MD Q2 | 19.59 | 7.58 | 18.28 | 9.77 | 1.31 | 5.31 | 0.010 | 0.005 | * |
| MD Q3 | 20.25 | 5.86 | 19.58 | 8.60 | 0.67 | 5.37 | 0.119 | 0.123 | * |
| MD Q4 | 19.06 | 7.61 | 18.00 | 10.08 | 1.06 | 5.37 | 0.018 | 0.006 | * |
| Duration | 3.32 | 1.07 | 6.40 | 1.43 | −3.07 | 1.66 | 0.0001 | 0.015 | * |

Q1 (superior nasal field); Q2 inferior nasal field); Q3 (superior temporal field); Q4 (inferior temporal field).
*Not corrected as study not powered for these comparisons.
MS, mean sensitivity; MD, mean deviation; SLV, standard loss variance; DD, diffuse defect.

## Sensitivity analysis: worst eyes only

Taking only the worst eye for each patient, results are presented for 126 patients (126 eyes) with 18 eyes at stage 3 and 108 at stage 4. The worst eye was right in 64 (50.8%) and left in 62 (49.2%).

### Size III versus size V differences

Results are outlined in table 2 for differences, and significance, for mean sensitivity, mean deviation and standard loss variance, along with differences across quadrants plus test duration. Differences for mean sensitivity between size III and V were about 6 dB overall (higher values for size V) and 5–7 dB across quadrants with greatest increases in inferior quadrants. A strong linear relationship and large correlation was found between mean sensitivity for size III and V stimuli (figure 3: $r^2$=0.678, p=0.0001). Using the loess curve to consider differences between size III and V stimuli, the point of greatest inflection (visualised on the scatterplot with value taken from y axis) is approximately 3–5 dB for size III stimulus.

Global mean deviation was significantly lower for size V but not clinically significant (figure 3) with significant differences only in inferior quadrants. A strong linear relationship and large correlation was found between mean deviation for size III and V stimuli (figure 3: $r^2$=0.687, p=0.0001). Using the loess curve to consider differences between size III and V stimuli, the point of greatest inflection is approximately 22–24 dB for size III stimulus. Diffuse defect was lower and standard loss variance was higher for size V (see online supplemental figure 2) representing more visual field responses. Test duration increased for size V but was likely due to more patient responses being obtained due to presence of more visible targets.

## DISCUSSION

Detection of visual field loss in glaucoma and other optic neuropathies is well reported with discussion of early detection methods, exploring visual field defects with steep border slopes and for conversion of disease/progression.[1 21] Studies have mostly used the 24–2 central static threshold programme.[3–5] In this study, we sought to explore the use of size V versus size III targets using the Octopus 900 G and LVC programmes to evaluate detection of absolute versus relative visual field loss in advanced (stages 3–4) glaucoma. This was a pragmatic clinical study in which we primarily wished to evaluate how much additional useful information could be obtained from a switch to stimulus size V at a stage of moderate to severe glaucoma when visual field results had reached a level of loss with stimulus size III, such that meaningful information could no longer be extracted from the results to guide clinical decision making.

One recent study sought to gather patient perspectives and reliability based on clinical grounds while comparing size III and V stimulus sizes using a 24–2 test programme.[22] They concluded that adjustment of testing to use of size V increased their patient cooperation and was a valid option when attempting to gather more reliable perimetry results. From our clinical perspective, we found the results using size V to provide more visual field data to help inform clinical decision making and a grounding towards future disease monitoring.

The mean sensitivity provides a single global indicator value for the overall visual field sensitivity. We found the mean sensitivity value to increase overall by about 6 dB for size V versus size III but with a difference of about 10 dB (in keeping with that expected by the change in

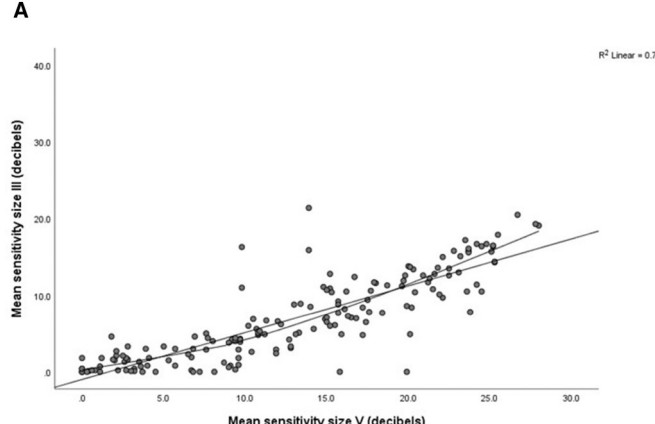

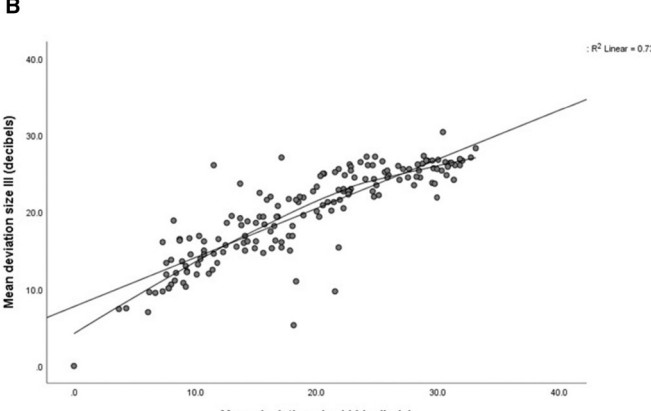

**Figure 2** Mean sensitivity and mean deviation comparisons—primary analysis of all eyes. (A) Mean sensitivity. Straight line is the linear regression line; curved line is the loess regression curve. Comparisons generally show clustering where mean sensitivity is close to 0 dB, that is, worst visual field. There is a 'floor' effect with size III, yielding many zeros, while size V gives, for most, larger values. The linear regression line and loess regression curve show large correlation for size V ($r^2$=0.731, p<0.0001). The loess point of inflection (value taken from the y axis) is approximately 4–6 dB. (B) Mean deviation. Straight line is the linear regression line; curved line is the loess regression curve. The linear regression line and loess regression curve show large correlation for size V ($r^2$=0.735, p<0.0001). The loess point of inflection (value taken from the y axis) is approximately 22–24 dB. loess= locally estimated scatterplot smoothing.

stimulus size) at mean sensitivities of 10–30 dB. This indicates less visual field loss for size V stimulus compared with the indication of absolute defects for size III stimulus when mean sensitivity values are low (<10 dB) and is indicative of the relationship between stimulus size and luminance.[22] Many perimeter parameters are such that at maximum luminance, the size III target has a calibrated range of 2–44 dB and the size V target has a range of 10–54 dB (10 dB difference). As expected, the larger stimulus size increases sensitivity and this is particularly so at greater eccentricities.[6] In clinical studies, this

is shown as the effective dynamic range of size V being about one log unit greater than size III.[3] Gardiner *et al* confirmed size V resulted in higher sensitivity at the same location, providing more reliable and less variable estimates of sensitivity in damaged visual fields. Their higher sensitivity was on average 5.6 dB—similar to our average of 6 dB and to the average of 6 and 7.6 dB reported by Morgan *et al*[22] and Choplin *et al*,[8] respectively.

The mean deviation provides a single global indicator value for the overall visual field sensitivity in comparison to an age-matched normal visual field. We found a slight but clinically insignificant reduction in mean deviation for size V vs III. The association between size III and V targets has been reported as approximately linear to a sensitivity of about 20 dB. Size V had a greater effective dynamic range with eight discriminable steps for progression—about twice as many as with size III.[23]

A second objective of our study was to consider indication levels for a transfer from size III to size V for static perimetry. In our analysis, we used the loess curve to compare differences between size III and V stimuli. The loess curve does not provide confidence values. We found the point of inflection to be within an overall range of approximately 4–6 dB for size III stimulus on mean sensitivity and 22–24 dB on mean deviation. Arguably, the inflection point might be considered the latest point at which to switch from size II to size V. A more sensible approach would be to switch before severe damage to the visual field occurs to allow progression to be more reliably tracked with size V over, potentially, a longer period of time. We therefore provide a clinical recommendation that at a mean sensitivity level of 10 dB (4 dB above the higher inflection estimate) and/or at a mean deviation level of 18 dB (4 dB below the lower inflection estimate) on a visual field using size III, a switch may be made to using size V stimuli for subsequent visual field assessment.

Mean deviation reduced more for inferior visual field quadrants. Our discrepancy between superior and inferior visual field cannot be explained by the number of test locations as test location numbers are equal for the superior and inferior visual field. The greater reduction in mean deviation for inferior visual field may be explained by the presence of better inferior visual field compared with the extent of superior visual field loss. In both primary open and closed angle glaucoma, visual field damage has often been reported as being more severe in the superior visual field than inferiorly.[24]

Variability for size V targets has been reported as being substantially lower than for size III.[3] However, in later studies, Wall and colleagues[4 5] did not find a significant reduction in size V variability compared with size III. This was partly explained by the use of size V allowing patients to detect more targets than with the smaller size III—0 dB responses using size III become detectable decibel responses with size V. Thus, as patients can see more, the extra responses add to variability. This was not apparent in cases with most severe visual field loss with mean sensitivities close to 0 dB but emerged from a mean sensitivity

**Table 2** Differences in visual field parameters for target size—worst eyes

| | G1, size III | | G1, size V | | Difference | | | |
|---|---|---|---|---|---|---|---|---|
| | Mean | SD | Mean | SD | Mean | SD | Paired T test | Bonferroni correction |
| MS | 6.31 | 5.57 | 12.24 | 7.67 | −5.93 | 4.43 | 0.0001 | 0.0002 |
| MD | 20.72 | 5.63 | 20.00 | 7.59 | 0.72 | 4.39 | 0.072 | 0.144 |
| SLV | 5.73 | 2.34 | 7.67 | 2.70 | −1.94 | 2.59 | 0.0001 | * |
| DD | 18.13 | 7.84 | 15.58 | 9.44 | 2.54 | 5.78 | 0.0001 | * |
| MS Q1 | 4.80 | 5.38 | 9.92 | 8.57 | −5.11 | 5.47 | 0.0001 | * |
| MS Q2 | 7.00 | 7.52 | 13.64 | 9.61 | −6.64 | 5.36 | 0.0001 | * |
| MS Q3 | 5.73 | 5.85 | 11.26 | 8.74 | −5.52 | 5.80 | 0.0001 | * |
| MS Q4 | 7.49 | 7.44 | 13.86 | 10.06 | −6.37 | 5.65 | 0.0001 | * |
| MD Q1 | 21.88 | 5.42 | 21.75 | 8.46 | 0.12 | 5.42 | 0.801 | * |
| MD Q2 | 20.34 | 7.57 | 19.08 | 9.60 | 1.27 | 5.34 | 0.01 | * |
| MD Q3 | 20.85 | 5.93 | 20.42 | 8.62 | 0.43 | 5.73 | 0.407 | * |
| MD Q4 | 19.85 | 7.46 | 18.86 | 9.99 | 0.99 | 5.64 | 0.053 | * |
| Duration | 3.31 | 1.07 | 6.36 | 1.37 | −3.05 | 1.65 | 0.0001 | * |

Q1 (superior nasal field); Q2 inferior nasal field); Q3 (superior temporal field); Q4 (inferior temporal field).
*Not corrected as study not powered for these comparisons.
MS, mean sensitivity; MD, mean deviation; SLV, standard loss variance; DD, diffuse defect.

of 10 dB or higher, that is, when sufficient visual field was discernible. We found mean deviation to be higher for size III generally up to 20–25 dB at which level mean deviation for size V was higher which likely reflects the greater extent of visual field seen with size V than size III. This effect disappeared for very poor visual fields, even with size V. Our study only included stages 3–4 advanced glaucoma with cases where there was substantial visual field loss to the extent that perimetry with size III no longer provided adequate information to facilitate monitoring of the condition for progression/response to treatment. Changing to size V allowed us to further evaluate visual fields with more visual field information being obtained.

The standard loss variance provides a single global indicator value for the extent of localised visual field loss that is present. Our results indicate higher standard loss variance measures with size V likely due to greater visual field responses being obtained with the larger size target. Further, and not surprising, we found longer test durations for size V compared with size III and in keeping with other studies.[22] This reflects the greater visual response to the larger size V target. Thus, the more targets than can be detected and responded to, the longer the test time will be.

We addressed a methodological issue with this paper. Many studies reporting visual field analysis present data from one eye: often the worst eye. We evaluated the distribution of data for all eyes in this study as our primary analysis. We subsequently ran a sensitivity analysis of worst eyes only to determine where significance lies when only focussing on the worst eye and where the sample size is smaller. We found no differences in distribution for right versus left eyes or for worst versus better eyes except

for mean deviation which showed more significance for primary analysis and which likely reflects the larger sample size for primary analysis (all eyes). Such evaluation of distribution is essential if wishing to combine results from all eyes in reporting data. Clearly analysis and reporting of data is important in the context of the type of study. Reporting intervention outcomes often necessitates a determination of independent effect. Hence reporting data from one eye per subject is often appropriate where the condition affects both eyes. However, in studies of assessment comparisons (such as this present study) the question being asked is about the ability of each assessment/test to detect the same defect and that question is as important for each eye individually (as two independent comparisons) as for the patients individually (as a single comparison). We have reported our methods and results in establishing the distribution of data and our subsequent decision to combine results versus solely presenting worst eye data only. As seen from our results across tables 1 and 2, values were highly comparable for all eyes versus worst eyes only.

### Limitations
Our study is limited by a number of factors. We report visual field results but did not collect visual acuity or fundus imaging for this study. We also did not formally record patient preference for visual field testing with size III or V. Further, we did not evaluate test reliability by accounting for false positive and false negative responses, although excessive responses were exclusion criteria for our study. We do not believe these are major limiting factors as we considered this a pragmatic clinical study centred on evaluating the differing visual field results

A

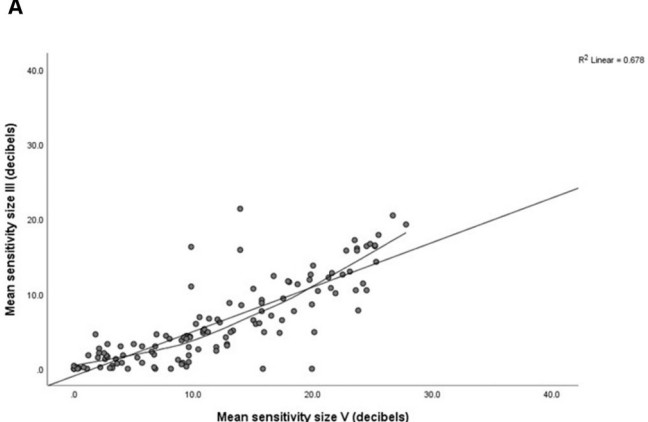

B

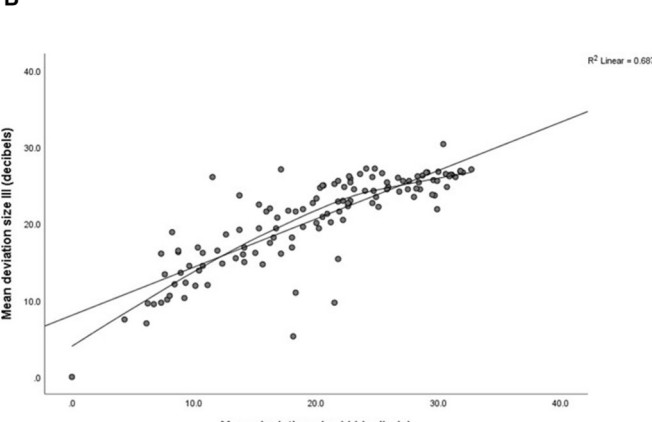

**Figure 3** Mean sensitivity and mean deviation comparisons—sensitivity analysis of worst eyes. (A) Mean sensitivity. straight line is the linear regression line; curved line is the loess regression curve. Comparisons generally show clustering where mean sensitivity is close to 0 dB, that is, worst visual field. There is a 'floor' effect with size III, yielding many zeros, while size V gives, for most, larger values. The linear regression line and loess regression curve show large correlation for size V ($r^2$=0.678, p<0.0001). The loess point of inflection (value taken from the y axis) is approximately 3–5 dB.(B) Mean deviation. Straight line is the linear regression line; curved line is the loess regression curve. The linear regression line and loess regression curve show large correlation for size V ($r^2$=0.687, p<0.0001). The loess point of inflection (value taken from the y axis) is approximately 22–24 dB. loess= locally estimated scatterplot smoothing.

obtained from size III and V stimuli. Hence, visual acuity and fundus imaging would not impact on our analysis. In future studies, we would wish to include patient feedback on preference for stimulus size as better patient engagement with visual field assessment is important.

## CONCLUSIONS

Use of size V versus III in standard automated perimetry for advanced stages 3–4 in glaucoma permits further visual field to be plotted in cases where the visual fields with size III no longer provides sufficient information

to appropriately monitor the disease. Mean sensitivity, standard loss variance and test duration increased in accordance with the extra visual field responses obtained using size V. We do not infer that automated perimetry with size V is better than with size III. Size V is easier to detect and, thus, from a pragmatic clinical view, a switch from size III to V in patients with advanced disease was a logical alteration to the visual field assessment protocol. We recommend considering a change to size V stimulus for advanced glaucoma at stages 3–4 when, with size III, the mean sensitivity reaches 10 dB and/or mean deviation reaches 18 dB, to facilitate continued assessment of visual fields as part of the clinical monitoring of the condition.

**Author affiliations**
[1]Ophthalmology, Glaucoma Clinic, New Delhi, India
[2]Glaucoma Service, Sadguru Netra Chikatsalya, Chitrakoot, India
[3]Head of Academics, Research and Training, Sadguru Netra Chikatsalya, Chitrakoot, India
[4]School of Computer Science and Mathematics, Liverpool John Moores University, Liverpool, UK
[5]Eye and Vision Science, University of Liverpool Faculty of Health and Life Sciences, Liverpool, UK
[6]Ophthalmology, SK Glaucoma Care Foundation, New Delhi, India
[7]Institute of Population Health, University of Liverpool Faculty of Health and Life Sciences, Liverpool, UK

**Contributors** FJR and DS conceived this study. DS and IS contributed to data collections. FJR and TS contributed to data extraction. FJR and GC contributed to data analysis. All authors contributed to writing this paper.

**Funding** The authors have not declared a specific grant for this research from any funding agency in the public, commercial or not-for-profit sectors.

**Competing interests** FJR reports consultancy work for Haag Streit AG and use of perimeters for research from Haag Streit AG and Zeiss Meditec, companies that may be affected by the research reported in the enclosed paper. There are no potential conflicts of interest reported by the other authors.

**Patient consent for publication** Not required.

**Ethics approval** Institution ethical approval (Ethics committee, Glaucoma Clinic New Delhi, India: Registration number GCND/EC/108) was obtained and patients provided informed consent.

**Provenance and peer review** Not commissioned; externally peer reviewed.

**Data availability statement** No data are available. The dataset for this paper is not available for sharing.

**ORCID iD**
Fiona J Rowe http://orcid.org/0000-0001-9210-9131

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
