## [Reviewer comments · BMJ Open]

ARTICLE DETAILS

TITLE (PROVISIONAL)	Standard automated perimetry using size III and size V stimuli in advanced stage glaucoma: an observational cross-sectional comparative study.
AUTHORS	Sood, Devindra; Czanner, Gabriella; Somerville, Tobi; Sood, Ishaana; Rowe, Fiona

VERSION 1 – REVIEW

REVIEWER	Wall, Michael University of Iowa
REVIEW RETURNED	30-Oct-2020

GENERAL COMMENTS	The authors report their experience comparing stimulus size III with stimulus size V in patients with advanced glaucoma in a prospective study. They found size V testing allowed continued monitoring of these patients that was not possible with size III due to its greater useful dynamic range. The study design and statistical analysis are appropriate and well thought out. The sample size is more than adequate. On page 4, the authors state there are potential issues when using size V for glaucoma patients with mild visual field damage. The evidence to date does not support this. In reference 5, when the data was binned by mean deviation, size V maintained its advantage to detect visual loss even in the bin with the best MD. Also, Flanagan JG, Artes PH, Wall M, et al. The Influence of Perimetric Stimulus Size on Defect Detectability in Early Glaucoma Invest. Ophthalmol. Vis. Sci. 2016; 58(8):2842, https://iovs.arvojournals.org/article.aspx?articleid=2557916 showed while size III testing resulted in a greater depth of the defect, size V because of its better retest variability flagged abnormal test locations in early glaucoma at least as well as size III.
---

REVIEWER	Abu, Sampson The University of Alabama at Birmingham
REVIEW RETURNED	03-May-2021

GENERAL COMMENTS	Standard automated perimetry comparisons (size III versus size V) in advanced stage glaucoma
--

	In a cross-sectional study to compare the use of size III and V in visual field testing among advanced glaucoma patients, Sood and colleagues report that, while sensitivity estimates obtained using both stimulus sizes have good correlation, testing with size V could provide additional useful clinical information when the dynamic range for size III becomes limited. The findings of this study are relevant to developing new strategies for effective monitoring advance glaucoma patients. I, however, have concerns for considerations. General comments: Could authors highlight which aspect of the study design makes this work a prospective study since patients were test once at one visit? What is clinical definition of glaucoma applied patients in your study population? Were study participants established glaucoma patients who were experienced test takers or were newly diagnosed? Comparative analysis is replicated with the worst eyes. Given the influence of intereye correlation, author should restrict analysis to inclusion of one eye per patient--randomly selecting one eye when two eyes are available. This will take care of duplicating the analyses with the worst eyes which produces similar results. In all the scatter plots, there seems to be a positive slope indicating the positive association between size III and V parameter. How the correlation (r) for mean deviation only tend is reported as negative. Could authors explain why? Also, for clarity and simplicity authors may chose report only r or the r2 value. And as matter of convention, p values are usually reported after the metric for association. The manuscript could also benefit from a professional proof-reading service. Specific comments: Author should rather use Q1, Q2 etc to indicate the four quadrants instead of the ST, IT, SN, IN sectors, else could be misconstrued for the commonly used which relatable to retinal orientation for structure-function relationships. They should indicate whether left eye field results were flipped in the orientation of the right eye since Q1 for LE corresponds to Q2 in RE.
--	---

	The G program is the test pattern. Which test strategy was used and could authors provide the additional information about the Octopus device used? E.g. manufacturers name, city etc. Include a legend in the scatter plots to distinguish between the two regression lines. Tables 1 and 2 have Bonferroni correction but it was not explained in the statistical analysis section which multiple comparisons are being adjusted for. Page 8, lines 24-25: “ For mean sensitivity sample size we assumed a difference of 3dB to be of clinical importance and hence to be detected.” Can the authors rephrase this sentence for clarity? Page 9, lines 12-14: “The difference in sensitivity was the dependent variables and the eye (right versus left) was treated as random factor nested within patient.” Consider rephrasing this statement for clarity. Is “-0.857” in page 14, line 32 the slope or correlation? Authors list a number of limitations. How do they affect the interpretation and generalizability of your findings? Page 16, line 20: "This indicates relative visual field loss for size V stimulus compared.....": Do you mean a loss for size III instead?
--	--

VERSION 1 – AUTHOR RESPONSE

Reviewer: Dr. Michael Wall, University of Iowa

The authors report their experience comparing stimulus size III with stimulus size V in patients with advanced glaucoma in a prospective study. They found size V testing allowed continued monitoring of these patients that was not possible with size III due to its greater useful dynamic range.

The study design and statistical analysis are appropriate and well thought out. The sample size is more than adequate.

A: Thank you for these positive comments.

On page 4, the authors state there are potential issues when using size V for glaucoma patients with mild visual field damage. The evidence to date does not support this. In reference 5, when the data was binned by mean deviation, size V maintained its advantage to detect visual loss even in the bin with the best MD. Also, Flanagan JG, Artes PH, Wall M, et al. The Influence of Perimetric Stimulus Size on Defect Detectability in Early Glaucoma Invest. Ophthalmol. Vis. Sci. 2016;

58(8):2842, <https://iovs.arvojournals.org/article.aspx?articleid=2557916> showed while size III testing

resulted in a greater depth of the defect, size V because of its better retest variability flagged abnormal test locations in early glaucoma at least as well as size III.

A: Thank you for highlighting this issue. We have imported the information provided into our Introduction.

Reviewer: Dr. Sampson Abu, The University of Alabama at Birmingham

In a cross-sectional study to compare the use of size III and V in visual field testing among advanced glaucoma patients, Sood and colleagues report that, while sensitivity estimates obtained using both stimulus sizes have good correlation, testing with size V could provide additional useful clinical information when the dynamic range for size III becomes limited. The findings of this study are relevant to developing new strategies for effective monitoring advance glaucoma patients.

A: Thank you for these positive comments.

1. Could authors highlight which aspect of the study design makes this work a prospective study since patients were test once at one visit?

A: We deemed this a prospective study in that the study was designed with pre-determined study testing protocol, outcome measures, eligibility criteria and recruitment process. It distinguishes this cross-sectional study from a retrospective cohort review.

2. What is clinical definition of glaucoma applied patients in your study population? Were study participants established glaucoma patients who were experienced test takers or were newly diagnosed?

A: We have clarified that patients were recruited with an existent clinical diagnosis of glaucoma. Glaucoma definitions and eligibility criteria are outlined in the methods.

3. Comparative analysis is replicated with the worst eyes. Given the influence of inter eye correlation, author should restrict analysis to inclusion of one eye per patient-randomly selecting one eye when two eyes are available. This will take care of duplicating the analyses with the worst eyes which produces similar results.

A: We have not amended the manuscript in relation to this comment. In our methods section and discussion section, we have specifically justified our intention to report all eye visual field data in addition to a further sensitivity analysis of worst eyes only. This tackles a methodological aspect of reporting visual field data – which we felt well-placed to address given the data available to us and the research team experience. With regard to our choice of worst eye, we have added a justification for this to the methods section.

4. In all the scatter plots, there seems to be a positive slope indicating the positive association between size III and V parameter. How the correlation (r) for mean deviation only tend is reported as negative. Could authors explain why?

A: The r values are now removed. The r² values are positive.

Also, for clarity and simplicity authors may chose report only r or the r² value.

A: Thank you for this suggestion. We have amended the results to report only the r² values.

And as matter of convention, p values are usually reported after the metric for association.

A: The p values have been moved.

5. The manuscript could also benefit from a professional proof-reading service.

A: The authors drafting the paper have English as their first language (UK resident). However, we acknowledge the differences between UK and US English presentation styles. We will defer to the journal editing style.

Specific comments:

1. Author should rather use Q1, Q2 etc to indicate the four quadrants instead of the ST, IT, SN, IN sectors, else could be misconstrued for the commonly used which relatable to retinal orientation for structure-function relationships. They should indicate whether left eye field results were flipped in the orientation of the right eye since Q1 for LE corresponds to Q2 in RE.

A: Thank you for this suggested. The manuscript has been amended accordingly.

2. The G program is the test pattern. Which test strategy was used and could authors provide the additional information about the Octopus device used? E.g. manufacturers name, city etc.

A: Amended to specify the G and LVC programmes with dynamic strategy. Manufacturer information has been added.

3. Include a legend in the scatter plots to distinguish between the two regression lines.

A: Legends have been provided separate to the figures – following the references. The first sentence provides information to distinguish between the two regression lines.

4. Tables 1 and 2 have Bonferroni correction but it was not explained in the statistical analysis section which multiple comparisons are being adjusted for.

A: This has been added.

5. Page 8, lines 24-25: “ For mean sensitivity sample size we assumed a difference of 3dB to be of clinical importance and hence to be detected.” Can the authors rephrase this sentence for clarity?

A: The sentence has been amended.

6. Page 9, lines 12-14: “The difference in sensitivity was the dependent variables and the eye (right versus left) was treated as random factor nested within patient.” Consider rephrasing this statement for clarity.

A: The sentence has been amended.

7. Is “-0.857” in page 14, line 32 the slope or correlation?

A: The correlation. This has been clarified in the text.

8. Authors list a number of limitations. How do they affect the interpretation and generalizability of your findings?

A: We have added discussion of our study limitations.

9. Page 16, line 20: "This indicates relative visual field loss for size V stimulus compared.....": Do you mean a loss for size III instead?

A: Relative means less for size V. We have amended the wording of this sentence.